# Antimicrobial Stewardship on Patients with Neutropenia: A Narrative Review Commissioned by Microorganisms

**DOI:** 10.3390/microorganisms11051127

**Published:** 2023-04-26

**Authors:** Joana Alves, Betânia Abreu, Pedro Palma, Emine Alp, Tarsila Vieceli, Jordi Rello

**Affiliations:** 1Infectious Diseases Department, Hospital de Braga, 4710-243 Braga, Portugal; joanamargaridaalves@gmail.com; 2Pharmaceuticals Department, Hospital de Braga, 4710-243 Braga, Portugal; betania.abreu.faria@hb.min-saude.pt; 3Infectious Diseases Department, Centro Hospitalar do Tâmega e Sousa, 4564-007 Penafiel, Portugal; pedropalmamartins@gmail.com; 4Infectious Diseases and Clinical Microbiology Department, Ankara Yıldırım Beyazıt University, 06760 Ankara, Turkey; eminealpmese@gmail.com; 5Infectious Diseases Department, Hospital de Clínicas de Porto Alegre, Porto Alegre 90035-903, Brazil; tvieceli@hcpa.edu.br; 6Clinical Research in Pneumonia & Sepsis (CRIPS), Vall d’Hebron Institute of Research (VHIR), 08035 Barcelona, Spain; 7FOREVA Research Pôle, Centre Hôpitalaire Universitaire de Nîmes, 30900 Nîmes, France

**Keywords:** immunocompromised patients, antimicrobial prescription optimization, critically ill patients, appropriate antibiotic, antibiotic dose adjustment

## Abstract

**Highlights:**

“Right first time” and achieving source control is crucial to optimize antibiotic management in neutropenic patients.The fear of missed pathogens and the risk of emergent resistant organisms require detailed knowledge of local patterns of susceptibility and a multidisciplinary team.A consistent pattern of safety requires shortening duration of therapy.Rapid diagnostic tools contribute to improve overall empiric antibiotic use, which should be a priority in neutropenic patients.A 5D approach is a core strategy to ensure improving antibiotic useEmphasis to a personalized prescription of antibiotics is required.

**Abstract:**

The emergence of antibiotic resistance poses a global health threat. High-risk patients such as those with neutropenia are particularly vulnerable to opportunistic infections, sepsis, and multidrug-resistant infections, and clinical outcomes remain the primary concern. Antimicrobial stewardship (AMS) programs should mainly focus on optimizing antibiotic use, decreasing adverse effects, and improving patient outcomes. There is a limited number of published studies assessing the impact of AMS programs on patients with neutropenia, where early appropriate antibiotic choice can be the difference between life and death. This narrative review updates the current advances in strategies of AMS for bacterial infections among high-risk patients with neutropenia. Diagnosis, drug, dose, duration, and de-escalation (5D) are the core variables among AMS strategies. Altered volumes of distribution can make standard dose regimens inadequate, and developing skills towards a personalized approach represents a major advance in therapy. Intensivists should partner antibiotic stewardship programs to improve patient care. Assembling multidisciplinary teams with trained and dedicated professionals for AMS is a priority.

## 1. Introduction

According to the World Health Organization (Geneva, Switzerland), the overuse and misuse of antibiotics is reflected in the emergence of resistance, posing a threat to global health security. Infections with multidrug-resistant (MDR) microorganisms lead to increased mortality, need for medical assistance, and treatment-related cost.

Infections are the most common cause of death in patients with malignancy [1]. Unfortunately, the emergence of MDR pathogens threatens the efficacy of treatment and the survival of patients that rely on antibiotics for treatment success. Patients with neutropenia are particularly vulnerable to infections; the risk of infection increases with severity and duration of neutropenia [2,3]. The incidence of febrile neutropenia varies according to malignancy. Patients with hematological malignancies have around 80% incidence compared to 10–50% in those receiving neoplastic therapy for solid tumors [4].

Since its first use in 1996 by John E. McGowan Jr. and Dale N. Gerding [5], the term “Antimicrobial Stewardship” (AMS) has become increasingly common in recent years. The objective of this definition is to highlight antimicrobials as a precious and nonrenewable resource, using a concept that encompasses rational use, and moving away from the concept of cost containment, which had prevailed until then. AMS programs enhance patient care by improving the use of antibiotic therapy, decreasing adverse effects, and improving patient outcomes [6,7,8,9,10,11]. There is a limited number of studies published regarding AMS programs on patients with neutropenia. This narrative review updates the current strategies of AMS applied to high-risk patients with neutropenia for bacterial infections, and suggests a five-point approach for optimizing antibiotic therapy in high-risk patients with neutropenia.

## 2. The Epidemiologic Change

Infections in patients with neutropenia are increasingly challenging, considering recent changes in the resistance patterns of microorganisms. Prolonged neutropenia subsequently leads to the prescription of extended and repeated courses of broad-spectrum antibiotics, contributing to selective pressure on susceptible bacteria and the increase of MDR organisms.

Patients with severe immunosuppression have damaged intestinal mucosal surfaces, which facilitates bacterial translocation across intestinal mucosa, resulting in potentially life-threatening infections [12,13].

Patients with malignancy are susceptible to develop infection, particularly when colonized with MDR pathogens [14]. Isolates from bloodstream infections in patients with hematological malignancies from recent studies are reported in Table 1.

Catheter-related, lung, and skin and soft tissue infections are among the most reported [15,16,17]. Gram-positive infections represent an important burden in these patients, particularly staphylococci [18]. Methicillin-resistance is common among coagulase-negative staphylococci and variable for S. aureus (19.2–80%) [15,17,19]; additionally, vancomycin intermediate *S. aureus* (VISA) and vancomycin resistant *Enterococcus* spp. may increase rates of treatment failure [20,21]. Cattaneo et al. reported a higher incidence of Gram-positive bloodstream infections during the induction phase of chemotherapy (where catheter-related infections may play a larger role), whereas Gram-negative infections were more common during the consolidation phase [15]. Moreover, enterococcal infections were more frequent in patients receiving antibiotic prophylaxis (9.7% vs. 2.6%, *p* = 0.016) [15]—a pathogen which is not adequately covered by quinolones, a class of antibiotics commonly used as prophylaxis. Infections due to penicillin-resistant *Streptococcus pneumoniae* have also been associated with higher mortality [22].

Of particular concern is the increasing number of invasive Gram-negative bacilli infections, which are associated with higher mortality rates [23]. Importantly, 30-day mortality is higher for MDR Gram-negative infections across studies, even when compared to infections due to resistant Gram-positive bacteria (such as MRSA and VRE) [15,17]. Beta-lactamase-producing Enterobacterales (ESBL-E) and MDR *Pseudomonas aeruginosa* are increasing; carbapenem-resistant *Acinetobacter baumanii* and carbapenemase-producing Enterobacterales (CRE) are spreading globally, with some centers reporting carbapenem-resistance rates of 10% in Enterobacterales, and up to 79% in nonfermenters [17,19].

**Table 1 microorganisms-11-01127-t001:** Epidemiology and outcomes of bloodstream infections in patients with hematological malignancies in European centers.

Reference	Country	Years	Study Type	Population	Pathogens	Resistance Type	30-Day Mortality
Cattaneo C et al., 2016 [15]	Italy	2012–2014	Prospective, multicentric	ALL and AML (n = 239, 433 BSI episodes)	GPB 44.8%: CoNS 25.4%, *S. aureus* 4.4%, *Enterococcus* spp. 2.6%	Methicillin-resistance: CoNS 77.6%, MRSA 80%	8.5%; higher for CRE + MDR-Pa: 29.6 vs. 6.1% (*p* < 0.01)
					GNB 38.4%: Enterobacterales 28.1%, *P. aeruginosa* 10.5%, Polymicrobial 15.7%, fungi 1.1%	ESBL-E 23.2%	
					During Induction phase: GBP 50.9%, GNB 31.9%, polymicrobial 13.8%, fungi 3.4%	CRE 9%	
					During consolidation phase: GPB 38.9%, GNB 46.8%, polymicrobial 14.3%, fungi 0%	MDR-Pa 27%	
Stoma I et al., 2016 [19]	Belarus	2013–2015	Prospective, single center	HSCT (n = 360, 135 BSI episodes)	GPB 34.9%: S. aureus 17%, CoNS 5.2%, Enterococci 3%	Methicillin-resistance: CoNS 27.8%, MRSA 65.2%	31.1%; higher for carbapenem-resistant nonfermenters (OR 5.46 (95% CI 1.33–20.7), *p* = 0.0126)
					GNB 64.4%: Enterobacterales 43.7%, Nonfermenters 21.5%	ESBL-E 40.7%, CRE 11.9%, Carbapenem-resistant nonfermenters 79.3%	
Scheich S et al., 2018 [16]	Germany	2008–2016	Retrospective, single center	HM w/GNB BSI (n = 109)	Only Gram-negative episodes evaluated: Enterobacterales 73.4%; Nonfermenters 26.6%	MDR pathogens * 19.4%: *P. aeruginosa* 37.5%, E. coli 34.4%, K. pneumoniae 25%	23.2%; higher for MDR vs. non-MDR GNB: 44.1% vs. 14.4% (*p* < 0.001)
Ali R et al., 2020 [17]	Turkey	2006–2016	Retrospective, single center	HM (n = 552, 950 BSI episodes)	GPB 48.3%: CoNS 37.8%, *S. aureus* 2.5%, *Enterococcus* spp. 5%	Methicillin-resistance: CoNS 84.6%, MRSA 19.2%	17.7%; higher for MDR-GNB infections: ESBL: 27.5% vs. 11.7% (*p* < 0.01), carbapenem-resistance: 65.4% vs. 14.8% (*p* < 0.01), MDR: 68.2% vs. 20.4% (*p* < 0.01)
						VRE 21.6%	
					GNB 42.4%: *E. coli* 19.4%, *Klebsiella* spp. 11.8%, *Pseudomonas* spp. 5.5%, *Acinetobacter baumanii* 3.5%	ESBL-E 39.4% CRE 9.8%	
						Carbapenem-resistant nonfermenters 32.6%	
Weber S et al., 2021 [24]	Germany	2006–2019	Retrospective, single center	HM, other hematological disorders, SOT (n = 391, 637 BSI episodes)	Common skin contaminants (coagulase-negative *Staphylococcus* spp., *Bacillus* spp., *Corynebacterium* spp., *Cutibacterium* spp., and *Micrococcus* spp.) 24.8%; *Escherichia* spp. 19%; *Enterococcus* spp. 13%	VRE 10%	Higher for carbapenem-resistant GNB infections: 62.5% vs. 4.7–18.7%
						MDR GNB 6.8%	
						Carbapenem-resistant GNB 2.5%	

Abbreviations: ALL, acute lymphoblastic leukemia; AML, acute myelogenous leukemia; BSI, bloodstream infection; CoNS, coagulase-negative staphylococci; CRE, carbapenem-resistant Enterobacterales; ESBL-E, extended spectrum beta-lactamase-producing Enterobacterales; GNB, Gram-negative bacteria; GPB, Gram-positive bacteria; HM, hematological malignancy; HSCT, hematopoietic stem cell transplantation; MDR, multidrug-resistant; MDR-Pa, multidrug-resistant *Pseudomonas aeruginosa*; SOT, solid organ transplant. * MDR was defined as resistance against at least three out of four antibiotic classes.

## 3. Antimicrobial Stewardship Programs in Patients with Neutropenia

AMS programs have several objectives, such as cost reduction, therapeutic optimization, and particularly, promoting actions that ultimately contribute to decreased antibiotic resistance rates [7].

In 2007, the Infectious Diseases Society of America (IDSA) defined AMS as ‘coordinated interventions designed to improve and measure the appropriate use of antimicrobial agents by promoting the selection of the optimal antimicrobial drug regimen including dosing, duration of therapy and route of administration’ [25]. In 2017, Dyar et al. [26] suggested that the term AMS could be defined as “A coherent set of actions which promote using antimicrobials responsibly”.

The literature on AMS in patients with malignancy is scarce by limited representation or exclusion of these patients in most studies, even though these patients are more vulnerable to MDR infections, and are more often exposed to broad-spectrum antibiotics. Antibiotic optimization is crucial to reduce morbidity and mortality rates in these patients. AMS general interventions such as formulary review, antibiotic restriction, and audit and feedback are effective in patients with malignancy [27,28,29,30].

A survey applied on solid-organ transplant centers in Switzerland exemplifies the lack of AMS implementation in immunosuppressed patients, with only a 29% response [31]. On the other hand, around 70% of intensive care units have an AMS program [32].

Most hospitals do not have dedicated staff working in antimicrobial stewardship [25,33], representing a gap that needs to be fulfilled. Health professionals working in this field must be well-trained and empowered to implement stewardship interventions on their hospitals. Therapeutic optimization is only achieved when a multidisciplinary team is involved. Multidisciplinary teams with infectious disease specialists, microbiologists, intensivists, oncologists, hematologists, pharmacists, and other front-line professionals are essential. It is important to involve nurses and pharmacists in strategies aiming to reduce unnecessary use of antibiotics [5,9]. AMS programs have a major role in minimizing the impact of multidrug-resistant bacteria, based on the implementation of a more conscious antibiotic prescription (e.g., according to local epidemiology, narrow spectrum, patient history) and the isolation of patients when in the presence of MDR bacteria.

## 4. AMS Strategies to Optimize Antibiotic Therapy

The emergence of resistance in pathogenic bacteria is a serious public health problem; still, prescribers misuse and over-use antibiotics [25,33,34,35,36]. It is difficult to change prescription behaviour; interventions such as antibiotic education, local clinical practice guidelines, audit and feedback, or antibiotic restriction have been shown to decrease antibiotic consumption in patients with febrile neutropenia without impacting the length of stay or mortality [27,28,29,30], and should be attempted. Which intervention to choose must be decided based on the local behaviour of prescribers.

This narrative review suggests a five-point approach for optimizing antibiotic therapy in high-risk patients with neutropenia (Table 2).

### 4.1. Diagnosis

The risk of development and the severity of infections are determined by a complex interplay between the pathogen and its virulence, the degree of impaired immunity, and of the host and the related cancer treatment.

Early diagnosis, at a stage when signs and symptoms may be absent and the site of infection may not be evident, is challenging. In patients with neutropenia undergoing chemotherapy, fever may be the only symptom that indicates bloodstream infection, which can result in severe sepsis, septic shock, and death [2,37].

#### 4.1.1. Risk Assessment

Antibiotic decision is based on risk assessment, and high-risk patients, with prolonged (>7 days duration), profound (<100 cells/mm^3^) neutropenia and/or comorbidities, should receive IV broad-spectrum empirical antibiotic treatment [37,38,39,40,41]. The risk assessment should also account for site of infection, clinical manifestations (such as hypotension), local epidemiological data, and history of previous infection/colonization by MDR microorganisms or previous antimicrobial use. Identifying pathogen carriage has an important role. Several studies report > 30% of any multidrug-resistant pathogen colonization in patients with hematological malignancies [16,42,43,44]; epidemiology varies between countries and centers. For instance, in an American study of 312 patients with acute myeloid leukemia, of which 40.9% were colonized with MDR organisms, 74.4% were vancomycin-resistant Enterococci (VRE), while ESBL-E and CRE represented 20% and 13.3%, respectively [42]. Conversely, in a Spanish study of 250 patients with hematological malignancies, of which 3.7% were colonized, most isolates were Gram-negative bacteria (89.9%), of which ESBL-E, carbapenem-resistant *Klebsiella* spp., and MDR *Pseudomonas aeruginosa* represented 24.1%, 0.7%, and 24.8%, respectively; Gram-positive bacteria accounted for 10.1% of isolates [44].

Importantly, some studies reported a considerable concordance between BSI isolates and colonization, as high as 80% [15,44]. Micozzi et al. compared the outcomes of BSI in patients with hematological malignancies who were KPC carriers between two periods (2012–2013 vs. 2017–2018): 68% of 27 patients in the first period developed BSI compared to 11% of 88 patients in the second period, following the use of empirical active antibiotic therapy against KPC (namely, ceftazidime–avibactam ± tigecycline and/or gentamycin) at the onset of febrile neutropenia. Similarly, inhospital mortality decreased (50% vs. 6%, *p* < 0.01) [45].

Screening for MDR organisms, particularly CRE and MRSA, as part of institutional policy provides early identification of carriers, and may help guide empirical antibiotic therapy decisions during febrile neutropenia episodes. The appropriate methodology and frequency of screening should be guided by local epidemiology, laboratory turnaround time, testing availability, and cost-effectiveness [46,47].

#### 4.1.2. Diagnostic Tools

When clinical diagnosis is difficult and complex, there is an increased risk of antimicrobial overuse. In this sense, the microbiology laboratory has a major role on AMS providing aetiological diagnosis. Morency-Potvin et al. highlighted the role of microbiology laboratories based on AMS strategies [48]. A close collaboration between microbiologist and the physician is the basis for a successful AMS implementation. Cumulative antimicrobial susceptibility reports, cascade or selective reporting, additional messages to enhance microbiology reports, rapid diagnostic testing, and rapid antimicrobial susceptibility testing are examples of diagnostic antimicrobial stewardship.

### 4.2. Conventional Microbiology

Conventional microbiology tests are the gold standard on diagnosis, yet may require up to 24 h for pathogen identification, and further 24–48 h for antibiotic susceptibility profiles. Timely identification and susceptibility results are the cornerstone for antibiotic optimization. Prescribers rely on susceptibility results; selective antimicrobial susceptibility reports should be used to promote the use of narrower-spectrum antibiotics [6,48].

### 4.3. Biomarkers

Biomarkers may help identify bacterial infections, limiting unnecessary antibiotic use. C-reactive protein and procalcitonin have been investigated as potential biomarkers for sepsis, considering that levels are increased in bacterial sepsis, but these should not be used alone to decide the diagnosis of bacterial infection [49,50,51]. Other biomarkers, such as proadrenomoduline, might add value to the diagnosis process to differentiate bacterial infection from a nonbacterial inflammatory response. The development of new therapies such, as T-cell CARS and TIL [52], emphasizes the importance of differentiating between the cytokine storm and true bacterial infection.

### 4.4. Rapid Diagnostic Testing and Rapid Antimicrobial Susceptibility Testing

Currently, the focus is on the use of rapid tests; results are obtained in a few hours with microorganism identification and susceptibility tests. Matrix-assisted laser desorption/ionization-time of flight (MALDI-TOF) mass spectrometry, usually used to identify pathogens [53], can be used for phenotypic resistance profiles, such as beta-lactam-resistance, from positive blood cultures [54]. VITEK-2 is an automated system for identification in around 3 h, and 18 h for susceptibility testing [55].

Other automated systems using a modified fluorescent in situ hybridization (FISH) assay for identification and microscopy for analysing bacterial growth rates and minimum inhibitory concentration (MIC) values extrapolated from positive blood cultures can provide results in <7 h [56].

Rapid phenotypic antimicrobial susceptibility testing (AST) is a recent EUCAST (European Committee on Antimicrobial Susceptibility Testing) method with an impact on AMS. Rapid AST improves antibiotic optimization in patients with Gram-negative sepsis without mortality differences [57]. Kim et al. [58] represented hematological patients in a randomized control trial, and MDR bacteremia had early optimal target antimicrobial therapy.

The constant change in the incidence of multidrug-resistance Gram-negative bacteria in patients with neutropenia requires rapid diagnostic tests for resistance identification; extended-spectrum B-lactamases and carbapenemases rapid identification, based on biochemical or fluorescence identification, have high sensitivities and specificities, are easy to perform, and provide results in 10–40 min [59,60]; prescribers will optimize antibiotics while waiting for the susceptibility test report.

### 4.5. Molecular Biology

Novel molecular assays for rapid diagnosis directly from blood samples may provide faster results for timely and pathogen-directed antibiotic initiation. Multiplex real-time PCR assays show high specificity, while sensitivity may vary (20–90%) [61,62]. However, these PCR assays are limited by the primers’ availability. The BioFireFilmArray BCID2 assay identifies 33 organisms by multiplex PCR from positive blood cultures in one hour, and detects 10 resistance markers [63,64].

Next generation sequencing (NGS) approaches allow for nearly universal identification without a priori knowledge of pathogens, and have demonstrated improved pathogen detection in septic patients, surpassing blood cultures [65]. Furthermore, molecular tests have the potential to identify pathogens in patients with previous exposure to antibiotics, which may inhibit growth in blood cultures.

The major drawback is susceptibility testing. Molecular detection of resistance is possible for Gram-positive bacteria from positive blood cultures, such as mecA and vanA/vanB gene identification for methicillin-resistance in staphylococcal and vancomycin-resistance in enterococcal infections, respectively [66,67]. Identifying susceptibility patterns in Gram-negative bacteria, which may present with multiple mechanisms of resistance, are more complex, and phenotypic resistance is not as easily predicted by available assays. Commercial array-based kits for extended-spectrum beta-lactamase and carbapenemase detection from positive blood cultures demonstrated high accuracy, but sensitivity and specificity were lower when compared to conventional methods, and may miss less frequent, yet important resistance phenotypes [68]. Even with the development of NGS, phenotypic identification of resistance must be a priority [69].

Pathogen-directed antibiotic therapy may be achieved, anticipating de-escalation. Still, currently these tests only complement conventional culture-based methods.

### 4.6. Drug

Choosing the right antibiotic for the right patient requires multidisciplinary collaboration, and is based on multiple factors between the patient, the pathogen, and the drug.

The impact of inappropriate antibiotic therapy in critically ill patients is well established: higher mortality, increased duration of hospital stays, and risk of drug resistance [70,71,72,73]. Each hour delay in administration of effective antibiotic increases the risk of mortality [74,75,76,77,78]. Patients with febrile neutropenia require rapid administration of empiric antibiotics [78,79].

In patients with neutropenia, the choice of the appropriate antibiotic (i.e., an antibiotic with in vitro activity against the infective pathogen) is of major importance, considering that these patients may not have typical warning signs and symptoms of infection; thus, empirical therapy is mostly based on broad-spectrum antibiotics.

The ultimate choice is based on local factors (hospital susceptibility patterns) and patient factors (previous hospital admission, previous antibiotic exposure, infection/colonization past history). Facility-specific clinical practice guidelines based on local microbiological data establish clear recommendations for optimal antibiotic use at the hospital.

Monotherapy has an equivalent efficacy compared with combination therapy [80,81,82]. Initial empiric treatment should include monotherapy with antipseudomonal beta-lactams [41,83], such as cefepime (fourth generation cephalosporin), piperacillin/tazobactam, or a carbapenem (meropenem or imipenem/cilastatin).

Due to the emergence of multidrug-resistant Gram-negative bacilli, ESBL-E, CRE, or MDR *Pseudomonas aeruginosa*, broad-spectrum antibiotics are justified for most patients, particularly high-risk patients with neutropenia [84,85,86,87,88].

Empirical therapy (Table 3) on febrile neutropenia usually does not cover MRSA unless suspected, since a meta-analysis of randomized control trials did not show advantages with vancomycin [89,90]; Gram-positive coverage with vancomycin (or other equal spectrum antibiotics) is suggested in patients with neutropenia. MRSA coverage is also recommended in suspected catheter-related infection, skin, or soft tissue infection, pneumonia, or septic shock. An interesting retrospective study [91] with 1305 patients with febrile neutropenia concluded that the use of empiric vancomycin is unnecessary in the majority of cases and did not improve mortality. MRSA screening is an important tool in antimicrobial stewardship, reducing the spectrum of empirical therapy [92,93].

Preventing drug adverse effects is a priority in patient safety. Therapeutic reconciliation is an effective strategy in preventing adverse drug reactions [94]. Medication reconciliation allows for the creation of the best possible list of all patient medication, including dose, frequency, and route of administration, based on several sources of information. After a complete drug history, omissions, duplications, or inadequate doses are analysed to prevent related incidents. It must be implemented whenever therapeutic changes or transition of care occurs [94,95].

The absence of medication reconciliation is responsible for 46% of medication-related errors, and more than 20% of the adverse effects that occur in hospitals [94]. Polypharmacy in patients with malignancies makes antibiotic prescription a challenge; the implementation of therapeutic reconciliation must be part of a stewardship program.

### 4.7. Dose

Adequate dosing and administration based on pharmacokinetics/pharmacodynamics (PK/PD) should be a priority. PK/PD describes the relationship between drug dose and pharmacological effects, with changes in drug concentrations leading to different pharmacological effects.

Increased volume of distribution requires a loading dose on hydrophilic antibiotics to achieve therapeutic concentrations in the target site [75,96,97]. Antibiotic properties should be taken into account, such as hydrophilic vs. lipophilic, bacteriostatic vs. bactericidal, time-dependent vs. dose-dependent.

Standard dosage regimens are based on healthy volunteers and do not take into account patient/pathogen characteristics. Underdosing of antibiotics is frequent in patients with sepsis, and is associated with treatment failures and worse outcomes [35,97]. Patients with severe infections are at higher risk of suboptimal antibiotic dose. High-risk neutropenic patients have increased volume of distribution and/or capillary leak syndrome and/or a more rapid clearance of certain drugs, particularly if mechanically ventilated or requiring vasopressor [96,97]. Higher doses are needed to obtain adequate serum concentrations.

The optimal dose (Table 4) includes the minimum inhibitory concentration (MIC) of the pathogen, and the site and the severity of the infection; additionally, the patient volume of distribution, renal clearance, weight, organ support (e.g., renal replacement therapy or extracorporeal membrane oxygenation) should be considered [96].

Prompt antibiotic administration and administration of a loading dose when indicated are associated with better patient outcomes in patients with septic shock, particularly within the first hour [75,76,77,78,96,97,98]. EUCAST has recently updated criteria for susceptibility interpretation [99]; time-dependent antibiotic efficacy depends on maintaining drug concentration above the pathogen MIC for longer than 40% of the time. Continuous prolonged infusion of beta-lactams increases efficacy and is associated with better outcomes [97,100,101].

A recent review [102] suggests a three-step approach to optimized antibiotic dose: the first antibiotic dose is based on pharmacodynamics, suggesting a double dose in critically ill patients with beta-lactams, aminoglycosides, and glycopeptides. Patients under continuous renal replacement therapy may also need increased doses from linezolid or colistin [102].

Therapeutic drug monitoring (TDM) is necessary to achieve the efficacy dose and to avoid overdosing, and consequently, adverse effects. TDM of glycopeptides and aminoglycosides has already been used for a long time; Bayesian statistics utilize measured concentration and clinical covariates, and they are the most accurate estimation of individualized patient dosing [100,103,104].

Vancomycin is the standard of care for Gram-positive infections due to MRSA. Inappropriate dosing is associated with high failure rates, higher MICs, and toxicity. Adequate vancomycin dosing focused on AUC/MIC improves clinical response in infections due to MRSA [105,106]. Most patients with neutropenia have augmented vancomycin clearance, and may need increased vancomycin daily dosing; TDM in patients with neutropenia is crucial to achieve an adequate antibiotic dose and to limit vancomycin dose variation depending on patient conditions [105].

Aminoglycosides dose adjustment based on TDM did not correlate with better outcomes, but contributed to limiting toxicity; yet, optimal dosing and earlier clinical response was achieved [104].

Beta-lactams are by far the most common used antibiotics and are the cornerstone of therapy of the majority of infections. Patients with neutropenia are a special population where beta-lactam TDM could lead to better outcomes, though their use is still controversial and further studies are needed to demonstrate their benefit [107]. Their efficacy is unpredictable since TDM is currently not recommended by practice guidelines, and only a few hospitals use it on a routine basis. Beta-lactam TDM provides real antibiotic measurements so that optimal dosing can be achieved. TDM is particularly important in critically ill patients where the optimal dose defines the outcome of the patients; there is an urgent call for better beta-lactam TDM.

### 4.8. Duration/Early Discontinuation

Overuse of antibiotics includes unnecessarily long duration of antibiotics. Long exposure to antibiotics leads to higher side effects, renal or hepatic toxicities, and bacterial resistance due to selective antimicrobial pressure. The traditional 7 to 14-days antibiotic course for most bacterial infections is outdated, and several randomized controlled trials comparing short course vs. traditional course provide evidence of equal efficacy, and fewer adverse effects and diminished emergence of resistance with short courses [108,109,110,111].

Discontinuation of empirical antibiotics after 72 h in hemodynamically stable patients with neutropenia in the absence of fever for the past 48 h is recommended by European Conference on infections in Leukemia (ECIL-4) guidelines [3], but is rarely applied on daily clinical practice.

Discontinuation of antibiotics in patients with persistent neutropenia who fill these criteria seems to be safe, and several studies demonstrated no substantial rates of infection recurrence [111,112,113,114].

A recent open-label, multicenter, randomized trial of 281 patients with febrile neutropenia who received intensive chemotherapy or hematopoietic stem-cell transplantation between 2014 and 2019 compared short (72 h) vs. extended (≥9 days until being afebrile for 5 days, or neutrophil recovery) carbapenem treatment. Early discontinuation was noninferior (10% margin) to extended treatment regarding treatment failure, both in intention-to-treat analysis (19% vs. 15%, adjusted risk difference 4.0% (90% CI: 1.7–9.5%, *p* = 0.25)) and per-protocol analysis (n = 225; 23% vs. 16%, adjusted risk difference 7.3% (90% CI: 0.3–14.9%, *p* = 0.11)). However, serious adverse events (16% vs. 10%) and all-cause mortality (3% vs. 1%) were higher in the short treatment group, driven by patients who are persistently febrile [115]; this supports the ECIL-4 [3] recommendation of discontinuation in stable and afebrile patients.

### 4.9. De-Escalation

De-escalation means implementing a strategy to reduce the spectrum of the initial empirical antibiotic therapy, either by isolating one valuable causal agent, or by excluding others. De-escalation is only manageable for routine bacteriology diagnosis, and contributes to reducing antibiotic exposure to broad-spectrum antibiotics and, so, may have a relevant impact on the emergence of resistance. European ICU practitioners regarded antibiotic de-escalation and discontinuation as the most important intervention, with the potential to prevent MDR development [116].

Martire et al. [117] implemented an AMS intervention in high-risk patients with neutropenia with de-escalation and discontinuation strategies, and a significant reduction of carbapenems consumption was achieved; rates of infection relapse, increase of ICU referral, and bacteremia were unchanged.

De-escalation appears to be safe and does not have a detriment impact on outcomes [118,119]. This is a strategy that needs further studies, especially to evaluate the impact on resistance, which is still an important intervention in AMS. Early discontinuation is safe, even in patients that remain febrile and neutropenic, only after exclusion of bacterial infection; this intervention demonstrated reduced antibiotic use without worse outcomes [118,120].

## 5. Conclusions

AMS is essential to ensure effective and safe access to antibiotics, regardless of the clinical situation. The scarcity of studies regarding AMS programs on patients with neutropenia limits this narrative review. Antimicrobial stewardship is expanding; there are still many constraints to overcome, such as the resistance of clinicians to limit antibiotic use, and the scarcity of the literature on improving outcomes. These gaps and additional evidence focusing on outcomes represent a research opportunity. There may be a conflict between the recommendations for the empirical prescription of antibiotics and AMS programs. These approach strategies highlight the importance of patient stratification to decide broad-spectrum antibiotics, the need for nonculture diagnostics tests for early diagnosis, the critical first empirical antibiotic dose, and the safety on de-escalation/discontinuation.

In patients with neutropenia, right first time is crucial. Screening and molecular diagnostic tests have an important role in early diagnosis or antibiotic discontinuation. Fear of attending clinicians of suboptimal coverage in high-risk neutropenic patients is a major issue. Empowering health professionals with educational activities and a multidisciplinary team with trained and dedicated professionals for antimicrobial stewardship is a priority.

## Figures and Tables

**Table 2 microorganisms-11-01127-t002:** Five-point approach for optimizing antibiotic treatment in critically ill patients with neutropenia.

Antimicrobial Stewardship Strategy	Rationale	Evidence Gaps
1. Diagnosis	The initiation of empirical antibiotic treatment should be prompted by fever and clinical signs, and not by C-reactive protein or other biomarkers.	
	Risk Assessment: - Patients with profound (<100 cell/mm^3^) and prolonged (>7 days) neutropenia have higher risk of infection and severity. - Screening for MDR pathogens (for ESBL-E, CRE, MRSA) may guide empirical antibiotics.	Appropriate screening methodology is unknown and dependent on local epidemiology.
	Early and specific microbiology diagnosis is essential to directed therapy and minimizing the use of broad-spectrum antibiotics.	Does not substitute conventional culture-based methods.
	Cascade or selective antimicrobial susceptibility reports promote available narrow-spectrum antibiotics.	No guidelines available.
	Rapid antimicrobial susceptibility testing improves management of antibiotic therapy in patients with Gram-negative sepsis.	Gram-positive infections.
2. Drug	Appropriate empirical antibiotic is associated with better patient outcomes.	
	Facility-specific treatment guidelines standardize prescribing practices based on local epidemiology.	
	Monotherapy versus combination therapy has equivalent efficacy.	
	Therapeutic reconciliation is an effective strategy in preventing adverse drug reactions.	
3. Dose	Underdosing of antibiotics is associated with treatment failures and worse outcomes.	
	Administer loading dose when indicated.	
	Prolonged infusion of beta-lactams is associated with better outcomes.	
	TDM achieves the efficacy dose and avoids overdosing and adverse effects.	Further studies are needed to demonstrate the benefit of beta-lactam TDM.
4. Duration (and early discontinuation)	Shorter antibiotic administration seems safe, does not have a detrimental impact on outcomes, and reduces the use of broad-spectrum antibiotics.	Further studies are needed to reinforce the evidence and to evaluate the impact on resistance.
5. De-escalation	De-escalation based on microbiology diagnosis is safe, does not have a detrimental impact on outcomes, and reduces the use of broad-spectrum antibiotics.	Further studies are needed to reinforce the evidence and to evaluate the impact on resistance.

PK/PD—pharmacokinetics/pharmacodynamics; TDM—therapeutic drug monitoring.

**Table 3 microorganisms-11-01127-t003:** Suggested empirical antibiotic treatment options for high-risk neutropenic patients *.

Local Epidemiology	Preferred Antibiotic Treatment	Alternative Antibiotic Treatment
Without risk for E-ESBL, CRE, MDR Pseudomonas aeruginosa, or MRSA	Cefepime or Piperacillin/tazobactam or Meropenem or Imipenem/cilastatin	Ceftazidime–avibactam or Aztreonam (consider addition of vancomycin for Gram-positive coverage) or Aminoglycoside (uncomplicated bloodstream infections with complete source control)
Presumed or confirmed extended-spectrum β-Lactamase-producing Enterobacterales	Meropenem or Imipenem/cilastatin	Ceftazidime–avibactam or Aztreonam (consider addition of vancomycin for Gram-positive coverage) or Aminoglycoside (uncomplicated bloodstream infections with complete source control)
Presumed or confirmed carbapenem-resistant Enterobacterales	Ceftazidime–avibactam or Meropenem–vaborbactam or Imipenem/cilastatin–relebactam	Cefiderocol or Aminoglycoside (uncomplicated bloodstream infections with complete source control)
Presumed or confirmed difficult-to-treat Pseudomonas aeruginosa	Ceftolozane–tazobactam or Ceftazidime–avibactam or Imipenem/cilastatin–relebactam	Cefiderocol or Aminoglycoside (uncomplicated bloodstream infections with complete source control)
Presumed or confirmed MRSA	Add vancomycin	Add Daptomycin or Linezolide

*** [2,3,41,83,84,85,86,87,88].

**Table 4 microorganisms-11-01127-t004:** Suggested dosing of antibiotics for the treatment of infections in high-risk patients with neutropenia.

Antibiotic	Adult Dosage, Assuming Normal Renal and Liver Function
Amikacin	20 mg/kg/dose IV (subsequent doses and dosing interval based on pharmacokinetic evaluation)
Aztreonam	2 g IV every 8 h, infused over 3 h
Cefepime	2 g IV every 8 h, infused over 3 h
Ceftazidime–avibactam	2.5 g IV every 8 h, infused over 3 h
Ceftolozane–tazobactam	3 g IV every 8 h, infused over 3 h
Gentamicin	7 mg/kg/dose IV (subsequent doses and dosing interval based on pharmacokinetic evaluation)
Imipenem/cilastatin	500 mg IV every 6 h, infused over 3 h
Imipenem/cilastatin–relebactam	1.25 g IV every 6 h, infused over 30 min
Meropenem	2 g IV every 8 h, infused over 3 h
Meropenem–vaborbactam	4 g IV every 8 h, infused over 3 h
Piperacillin-tazobactam	4.5 g IV every 6 h, infused over 3 h
Plazomicin	15 mg/kg/dose IV (subsequent doses and dosing interval based on pharmacokinetic evaluation)
Vancomycin	Loading dose up to 35 mg/kg IV (maximum 3 g) and maintenance dose 15–20 mg/kg 8–12 h, adjusted to achieve target AUC24 of 400–600 (subsequent doses based on pharmacokinetic evaluation)

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
