# Peer review of "Antimicrobial Stewardship on Patients with Neutropenia: A Narrative Review Commissioned by Microorganisms"

_microorganisms, 2023, doi:10.3390/microorganisms11051127_

Round 1

Reviewer 1 Report

The chosen subject is of great relevance and importance in medical practice. I recommend the continuation and implementation of the discussed aspects in practice.

This review updates the current strategies of AMS applied to high-risk patients with neutropenia for bacterial infections. Antimicrobial stewardship programs in patients with neutropenia is relevant in the field of infection therapy.  Infections in patients with neutropenia are increasingly challenging, considering the changes in the bacteria resistance patterns to the antibiotics. This narrative review suggests a five-point approach for optimizing antibiotic therapy in high-risk patients with neutropenia. the authors should consider: rapid tests have an important role on early diagnosis or antibiotic: Novel molecular assays for diagnosis directly from blood samples may provide faster results for timely and pathogen directed antibiotic initiation. The introduction of automatic Vitek method is desirable.

These approach strategies highlight the importance of patient stratification to decide broad-spectrum antibiotics, the need for early diagnosis, the critical first empirical antibiotic dose and the safety on de-escalation/discontinuation.

The tables contain clearly presented and comprehensive data. I recommend a re-arrangement of the tables according to the classic model. Figure 1.The figure is rather presentation of the data (Interactions between the host, pathogen and drug) in 3 colored columns. The data can be presented in a single table.

Author Response

Thank you so much for your comment. We added:

  • Line 307: VITEK-2 is an automated system for identification in around 3 hours and 18 hours for susceptibility testing (55).
  • Line 332: The BioFireFilmArray BCID2 assay identifies 33 organisms by multiplex PCR from positive blood cultures in one hour and detects 10 resistance markers (63,64)

Regarding figure 1, in light of another reviewer, we decided to eliminate figure 1.

Reviewer 2 Report

Authors reported an interesting review about the role of AMS in neutropenic patients. The topic is crucial and we need appropriate literature.

The manuscript is well-written, references are appropriate and updated. Recommendations are clear and based on scientific data.

I suggest only to remove Figure 1, then the manuscript should be accepted.

Author Response

Thank you so much for your words. We agree and decided to eliminate figure 1.

Reviewer 3 Report

General comments

=============

I appreciated the opportunity to peer-review your work on antimicrobial stewardship on patients with neutropenia. This manuscript was well written. 

Specific comments

=============

Major comments

---------------------

1. Please add an overview for febrile neutropenia with each background (malignancy, hematological malignancy, and transplantation) in the introduction.

2. Please add the future direction of AMS in febrile neutropenia.

3. Please add the limitation of this narrative review.

Minor comments

---------------------

4. Please add the reference numbers in table1.

5. Please add the explanation for the abbreviation of MIC in figure1, ECIL-4 (line 365), EUCAST.

6. Please change the “Alergies” to “Allergies” in FIgure 1.

Author Response

Major comments

  1.  Thank you so much for this comment to value this manuscript  We added “The incidence of febrile neutropenia varies according to malignancy. Patients with hematological malignancies have around 80% incidence compared to 10-50% in those receiving neoplastic therapy for solid tumors (4).”
  2. We added line 116: “…and suggests a five-point approach for optimizing antibiotic therapy in high-risk patients with neutropenia.
  1. We added Line 621: “The scarcity of studies regarding AMS programs on patients with neutropenia limits this narrative review.”

Minor comments

5. Thank you for your comment. References were added.

6. Thank you for your comment. Abbreviations were described.

7. In light of previous reviewers, we decided to eliminate figure 1.